# Ultraviolet-C-Based Mobile Phone Sanitisation for Global Public Health and Infection Control

**DOI:** 10.3390/microorganisms11081876

**Published:** 2023-07-25

**Authors:** Matthew Olsen, Adrian Goldsworthy, Rania Nassar, Abiola Senok, Abdullah Albastaki, Zheng Z. Lee, Sam Abraham, Rashed Alghafri, Lotti Tajouri, Simon McKirdy

**Affiliations:** 1Faculty of Health Sciences and Medicine, Bond University, Robina, QLD 4226, Australia; 2College of Medicine, Mohammed Bin Rashid University of Medicine and Health Sciences, Dubai P.O. Box 505055, United Arab Emirates; 3Oral and Biomedical Sciences, School of Dentistry, College of Biomedical and Life Sciences, Cardiff University, Cardiff CF10 3AT, UK; 4Dubai Police Scientists Council, Dubai Police, Dubai P.O. Box 1493, United Arab Emirates; 5General Department of Forensic Sciences and Criminology, Dubai Police, Dubai P.O. Box 1493, United Arab Emirates; 6Harry Butler Institute, Murdoch University, Murdoch, WA 6150, Australia

**Keywords:** pathogenic, public health, antimicrobial, disinfect

## Abstract

Introduction. Mobile phones act as fomites that pose a global public health risk of disseminating microorganisms, including highly pathogenic strains possessing antimicrobial resistances. The use of ultraviolet-C (UV-C) to sanitise mobile phones presents an alternative means to complement basic hand hygiene to prevent the cross-contamination and dissemination of microorganisms between hands and mobile phones. Aim. This study aimed to evaluate the germicidal efficacy of the Glissner CleanPhone UV-C phone sanitiser (Glissner) device. Methods. Two experimental trials were performed for the evaluation of the CleanPhone (Glissner). The first was a controlled trial, where the germicidal efficacy of the CleanPhone was evaluated against six different microorganism species that were inoculated onto mobile phones. The second was a field trial evaluating the germicidal efficacy of the CleanPhone on 100 volunteer mobile phones. Efficacy was determined based on colony counts of microorganisms on Columbia sheep blood agar before and after UV-C treatment. Results. In the controlled trial, reduction in growth was observed for all microorganisms after UV-C treatment with ST131 *Escherichia coli* showing the highest growth reduction at 4 log_10_ CFU/mL followed by *C. albicans* and ATCC *E. coli* at 3 log_10_ CFU/mL. An overall reduction in microorganism growth after UV-C treatment was also observed for the field trial, with an average growth reduction of 84.4% and 93.6% in colony counts at 24 h and 48 h post-incubation, respectively. Conclusion. The findings demonstrated the capability of the CleanPhone (Glissner) to rapidly sanitise mobile phones, thereby providing a means to reduce the potential dissemination of microorganisms, including highly pathogenic strains with antimicrobial resistance.

## 1. Introduction

Mobile phones are everywhere and owned by billions of individuals worldwide. Current 2023 estimations show that the sale and ownership of these devices are at 6.84 billion and will grow to 7.69 billion by 2027 [1]. However, a recent scoping study reviewed 56 scientific publications, sourced from 24 different countries, and identified that mobile phones are important fomites in health care settings [2] and within the community [3,4]. Mobile phones are reported to possibly play the role of ‘Trojan Horse’ fomite platforms, never or rarely decontaminated [3]. The extent of microbes found on contaminated mobile phones have been reported in recent studies with high throughput metagenomic DNA sequencing [2,4,5,6]. Olsen et al., 2022, found more than 11 thousand microbes, including fungi, protozoa, bacteria, and viruses across 26 mobile phones belonging to paediatric medical staff personnel [5]. In that study, 56% of health care workers admitted to using mobile phones in the bathroom. The use of mobile phones in bathrooms is a common practice around the globe [7,8] and poses real risks to global public health, especially in the medical, elderly/childcare, aviation/biosecurity, and food-handling industries.

In our daily activities, our two hands are in contact with a plethora of fomites, and touching our phones constantly leads to a dynamic and ongoing cross contamination of mobile phones with the deposition of thousands of microbes. It is also likely that the high microbial load on mobile phones would cross-contaminate our hands even after handwashing. Mobile phones are used on average for 3–5 h [9] a day and have become extensions of our two hands like a third hand [5] that we rarely ‘clean’. While the World Health Organisation (WHO) and other public health authorities promote active hand hygiene campaigns all around the world, mobile phones harbouring microbes jeopardise and negate hand hygiene. Viruses like severe acute respiratory syndrome coronavirus-2 (SARS-CoV-2) has been detected on mobile phones [6,10,11,12] and found to remain on mobile phones for up to 28 days [13], leading to the hypothesis that the use of mobile phones plays a role in the dissemination of the virus [3].

To address this public health concern, prevention measures to decontaminate mobile phones have been released by some phone manufacturers with the use of alcohol-based wipes. While easily accessible, wipes pose several inconvenient issues by being abrasive to phone surfaces [14], suboptimal surface coverage when wiping down the devices, and generating unsustainable large waste of used wipes globally. Additionally, adoption and compliance of this mode of decontamination might be suboptimal.

A novel alternative is the use of ultraviolet-C (UV-C) sanitisation of phones to overcome these limitations. UV-C emitting technologies are based on lamps or LED based sources of approximately 265 nm germicidal waves that are directed onto surfaces for sanitisation and are widely used in medical laboratories worldwide. The intensity of such UV-C emission dictates the dose that is applied onto a given surface. The germicidal wavelength of the UV-C is then absorbed in the DNA/RNA of microorganisms, leading to genomic breakdown and death of these microorganisms. Ideally, UV-C phone sanitisation could be coupled with hand hygiene to prevent the dynamic cross-contamination between hands and mobile phones (or vice versa).

With mobile phones circulating in billions, little attention is given to their fomite nature posing a risk for microbial dissemination globally. Practical UV-C phone sanitisers may be the solution to decontaminate such mobile phones. Several UV-C phone sanitisers are available in the market, with few studies evaluating their strengths and limitations. Practically speaking, phone UV-C sanitisers should be rapid (less than 20 s), of high germicidal efficacy, designed as a hands-free experience, ergonomic to be used anytime/everywhere, certified, and entirely safe (enclosed UV-C emission). The aim of this study is, therefore, to evaluate the germicidal capacity of the CleanPhone UV-C phone sanitiser (Glissner) by applying a ten second UV-C treatment on mobile phones.

## 2. Materials and Methods

### Specification of the CleanPhone Ultraviolet-C Phone Sanitiser (Glissner)

The Glissner CleanPhone UV-C sanitiser is an enclosed apparatus with germicidal LED emitting UV-C at a 265–275 nm wavelength range with a power of 50–60 mJ/cm^2^. The UV-C lamp consists of 48 LEDs with two lamps (so 96 LEDs in total) enclosed within the CleanPhone, which enables a 360° sanitisation coverage.

Experiment A: Controlled trial. The performance of the CleanPhone (Glissner) was evaluated by observing colony counts on mobile phones inoculated with selected microorganisms before and after UV-C light-based sterilisation treatments with an overview of the general procedure shown in Figure 1.

One fungal and six bacterial strains representing species commonly isolated from humans as commensals or pathogens were selected. ATCC 90028 *Candida albicans* (*C. albicans*) was the selected fungal strain and, together with the two bacterial strains ATCC 1228 *Staphylococcus epidermidis* (*S. epidermidis*) and ATCC 25922 *Escherichia coli* (ATCC *E. coli*), are strains commonly used in medical laboratories as quality control strains. The remaining four bacterial strains were archival in-house strains of species known to be pathogenic to humans and/or possess antimicrobial resistance that were vancomycin-resistant, *Enterococcus* (VRE), ST131 *E. coli* (ST131), *Salmonella*, and methicillin-resistant *Staphylococcus aureus* (MRSA). Both ST131 and MRSA were isolated from pigs, whereas *Salmonella* and VRE were isolated from cattle and humans, respectively. Prior to the commencement of experiment, fresh inoculum of each microorganism was made. One pure colony of each bacterial strain was inoculated into 10 mL of brain heart infusion broth (BD Difco^TM^, Thermo Fisher Scientific) and incubated at 37 °C for 16 to 20 h, whereas one pure colony of the fungal strain was inoculated into 10 mL of Roswell Park Memorial Institute (RPMI) broth (Gibco^TM^, Thermo Fisher Scientific, Waltham, MA, USA) supplemented with 2% glucose (*w*/*v*) and buffered with 0.165 morpholine propanesulfonic acid to pH 7.5 and incubated at 37 °C for up to 48 h. A consistent template for microorganism inoculation and swabbing was also created for the mobile phones by putting the smallest mobile phone onto each phone and putting stickers on the screen of the larger mobile phone (bottom phone) and around the smaller mobile phone (top phone), thereby creating a consistent template area based on the smallest mobile phone.

Prior to inoculation, each mobile phone was sterilised by spraying the screen with 70% ethanol followed by wiping the screen dry and inserting the mobile phone into the CleanPhone (Glissner) for ten seconds (Figure 1A). After sterilisation, 75 µL of one incubated bacterium inoculum was dispensed within the template area of the mobile phone and spread evenly across the template area using a sterile loop. This was repeated five more times on five different mobile phones for a total of six mobile phone replicates per inoculum (Figure 1B). The inoculum on each mobile phone was then air dried until the liquid within the template area has evaporated. Once dried, three mobile phones were randomly selected for sterilisation treatment with the CleanPhone (UV-C-light-treated phones), where each mobile phone was inserted into the sanitiser for ten seconds while the remaining three were not sterilised (UV-C light non-treated phones). A sterile cotton swab was dipped into sterile 1× phosphate-buffered saline (PBS) and used to swab the template area of each mobile phone. The swab was then placed into a 2 mL tube containing 1 mL of sterile 1× PBS to await further processing. The experiment was repeated (including the pre-inoculation sterilisation procedure) for each remaining inoculum on the same day.

All tubes were vortexed and underwent 10-fold serial dilution to 10^−5^ in sterile 1× PBS. UV-C light non-treated phones were inoculated onto Columbia sheep blood agar (SBA) (Edwards Group) at neat to 10^−5^, whereas UV-C-light-treated phones were inoculated at neat to 10^−3^. Inoculation was performed by dispensing 75 µL of the corresponding diluted inoculum onto the agar and spread evenly across the agar surface using a sterile loop. All agars inoculated with the six bacterial strains were incubated at 37 °C for 16 to 20 h, whereas agars inoculated with *C. albicans* were incubated at 37 °C for up to 48 h. Colony counts were performed on all agars after incubation with the concentration of microorganism growth on agars expressed in colony-forming units per mL (CFU/mL) of inoculated PBS.

Experiment B: Field trial evaluation. The performance of the CleanPhone (Glissner) was evaluated with a field trial involving 100 mobile phones acquired from volunteers with an overview of the general procedure shown in Figure 2. Each mobile phone was divided into left and right sections by placing stickers vertically through the centre of the mobile phone screen. A sterile cotton swab was dipped into sterile water and used to swab the left section of a mobile phone screen followed by immediate inoculation onto SBA using the lawn spread technique (before treatment agar). The mobile phone was then inserted into the sanitiser for ten seconds. The swabbing and agar inoculation procedure was repeated on the right section of the treated mobile phone screen with a sterile cotton swab (after treatment agar). All agars were incubated at 37 °C for 24 h with colony counts performed on all agars after incubation (24 h post-incubation). All agars were further incubated at 37 °C for another 24 h followed by another round of colony counts (48 h post-incubation) (Figure 2).

Data analysis. Digital imaging of colonies on agars were all electronically captured for processing and descriptive analysis in the Stata analysis package (version 16.0, Stata Corporation, TX, USA). Quantitative counts of microorganism growth in CFU/mL of PBS were log_10_ transformed where required for interpretation and analysis. For Experiment A, a one-way analysis of variance (ANOVA) followed by a post hoc Tukey test was performed for comparing significant differences in log_10_ CFU/mL between treated and non-treated phones and between the six micro-organism strains. The same statistical analysis was also performed for Experiment B for comparing significant differences in colony counts between treated and non-treated phones at 24 h and 48 h post-incubation.

## 3. Results

Experiment A: Comparison of microorganism growth between inoculated untreated and treated mobile phones. Growth was observed on all agars with microorganism growth ranging from 4 to 7 log_10_ CFU/mL on the UV-C light non-treated phones and 1 to 4 log_10_ CFU/mL on UV-C-light-treated phones (Figure 3). For the UV-C light non-treated phones, *C. albicans* had the lowest average growth at 4 log_10_ CFU/mL, whereas ST131 and ATCC *E. coli* had the highest average growth at 7 log_10_ CFU/mL. For UV-C-light-treated mobile phones, *C. albicans* also had the lowest average growth at 1 log_10_ CFU/mL, followed by ST131 with an average growth of 3 log_10_ CFU/mL. The remaining five microorganisms all had an average growth of 4 log_10_ CFU/mL for the UV-C-light-treated phones. Thus, despite all microorganisms having an average reduction of at least 2 log_10_ CFU/mL in growth concentration between untreated and treated mobile phones, ST131 had the greatest difference at 4 log_10_ CFU/mL followed by *C. albicans* and ATCC *E. coli* at 3 log_10_ CFU/mL. No significant difference was found in colony counts between micro-organisms, though the differences in colony counts between treated and non-treated phones were found to be significant (*p* < 0.001).

Experiment B: Comparison of microorganism growth on treated and untreated mobile phones at 24 h and 48 h post-incubation. Colony counts on mobile phones before and after sterilisation treatment with the CleanPhone Glissner were compared at 24 h and 48 h post-incubation. At 24 h post-incubation, growth was observed on 79 mobile phones on before treatment agars with an average colony count of 27.68 (total colony count = 2768), although this was reduced to 15 mobile phones, with an average colony count of 1.56 (total colony count = 156) after treatment (Figure 4a). However, it was observed that two mobile phones had colony counts on the after-treatment agars that were higher than the before-treatment agars. At 48 h post-incubation, the number of mobile phones with growth on before treatment agars increased to 93, with an average colony count of 47.89 (total colony count = 4789), whereas growth on after treatment agars increased to 42 mobile phones although with a reduced average colony count of 3.07 (total colony count = 307). Overall, this showed an average reduction of 94.4% in colony counts at 24 h post-incubation and an average reduction of 93.6% at 48 h post-incubation (Figure 4b,c). At 24 h and 48 h post-incubation, significant differences in colony counts were found between treated and non-treated phones (*p* < 0.01 and *p* < 0.001, respectively).

Figure 4a–c Colony counting of swabs obtained from UV-C- and non-UV-C-treated mobile phones.

## 4. Discussion

Mobile phones are platforms that are subject to a dynamic and constant contamination and deposition of microorganisms [3], with the added concern of mobile phones acting as fomites in healthcare settings, thereby posing high risk for nosocomial diseases and dissemination of dangerous pathogens to immunocompromised individuals and patients [2,3,4,5,6,7,8]. The effectiveness of the CleanPhone (Glissner), a UV-C mobile phone sanitiser designed for placements in offices, healthcare, and biosecurity sites to reduce the microbial load of microorganisms on mobile phones was evaluated.

While the CleanPhone (Glissner) was not able to eliminate all microorganisms inoculated onto mobile phones, the device was still able to achieve a minimum reduction in microbial load of at least 2 log_10_ CFU/mL in ten seconds (at least100 times less microbial burden in 10 s). The absence of any significant difference between the micro-organisms used in this study further indicates that the CleanPhone is equally capable of eliminating all tested micro-organisms. Considering that the tested microorganisms include ST131 and MRSA, this reduction would be significant, as the former is reported globally as a widely disseminated and highly pathogenic *E. coli* strain with resistance towards multiple antimicrobials [15,16,17], whereas the latter is a multi-drug resistant pathogen reported globally to be responsible for nosocomial outbreaks [18,19]. The use of the CleanPhone (Glissner) in key strategic locations such as healthcare facilities, nursing homes, and hospitals may assist in reducing the dissemination of these dangerous bacteria strains, especially with ST131, which had the highest reduction in microbial load.

The CleanPhone (Glissner) was also found to be effective in reducing the microbial load of the fungi *C. albicans*. A recent study reported that 32% of mobile phones were contaminated with fungi, with the most frequent isolated species being *C. albicans* [20]. With the WHO recently categorising *C. albicans* as a critical priority threat towards public health [21], the use of the CleanPhone (Glissner) at key strategic locations would also make the device an important tool in improving public health. Moreover, several other *Candida* species were also designated by WHO as critical or high priority threat. While further research and evaluation would be required to confirm the CleanPhone’s effectiveness against other *Candida* species (and other pathogenic fungi), its current effectiveness against *C. albicans* makes it a highly potential device in combating the spread of these dangerous fungal species.

When tested on mobile phones through a comprehensive field trial, the CleanPhone (Glissner) reduced the overall microbial load across 100 mobile phones from average colony counts of 27.68 and 47.89 at 24 h and 48 h post-incubation, respectively, to 1.56 and 3.07 colonies at 24 h and 48 h post-incubation, respectively. These translated to consistent 95% and 94% reductions in microbial loads at 24 h and 48 h post-incubation, respectively. These significant reductions for both post-incubation observations indicate that the CleanPhone (Glissner) was equally effective in reducing fast- (growth within 24 h of incubation) and slow-growing (growth within 48 h incubation) microorganisms, thereby highlighting the CleanPhone’s versatility against different microorganism species present on public mobile phones. However, two limitations were present in this study. Firstly, evaluation was limited to the screen area of the mobile phone; thus, it is unknown how effective would the CleanPhone (Glissner) be in sanitising other areas of mobile phones. Secondly, the effectiveness of the CleanPhone (Glissner) against viruses was not evaluated, and its effectiveness against fungi was only performed on one species. Further investigation would thus be necessary to further improve the capabilities of the Glissner’s UV-C lighting technology.

Other studies have looked at other UV-C light phone-sanitising technologies. In 2018, one of these studies reported that UV-C devices were better compared to the usage of wipes or disinfectants on aerobic bacteria [22]. However, these two UV-C devices required 2 and 5 min UV-C long exposure time, respectively (60 mJ/cm^2^ with irradiances of 500 and 200 µWcm^2^), and were most likely impractical for healthcare staff to use during patient care. In 2020, a study reported that 30 s and 1 min UV-C light exposure to 30 mobile phones belonging to health care staff resulted in 90.5% and 99.9% colony-forming unit reductions, respectively [23]. Another study performed in a Netherlands clinical healthcare evaluated the UV-SmartD25, a 25 kg UV-C light technology, on 100 smartphones, 100 digital European cordless telecommunication phones, and 100 ViSi Mobiles, resulting, after 30 s UV-C light exposure, in an overall mean of colony-forming unit reduction of 97.9% [24]. However, these devices require users to physically open, close, and reopen the chamber’s door that ultimately favour or prone for cross-contamination of their hands when operating these sanitisers.

Despite the limitations, results of our current study show the potential of the CleanPhone (Glissner) in providing a means to reduce potential dissemination of highly pathogenic and resistant bacteria and fungi strains through rapid (10 s) and effective sanitisation of mobile phones at locations with high public access such as airports, tourist attractions, and offices or at locations with vulnerable and immunocompromised people such as nursing homes and hospitals. Additionally, the device would further complement personal mobile phone sanitisers currently available to the public and basic hand hygiene to further reduce microbial load of mobile phones. Further evaluation of the CleanPhone’s effectiveness against viruses, other pathogenic fungi species, and its sanitisation effectiveness throughout the whole mobile phone would further improve the device’s capabilities to improve public health.

## 5. Conclusions

Mobile phones, often regarded as our ‘third hands’, are important contaminated platforms, harbouring all classes of microbes (viruses, bacteria, fungi, and protozoa), with also the presence of antimicrobial-resistant microbes. Mobile phones are negating hand washing, a WHO life-saving campaign, are numbered in billions worldwide, and are crossing borders with the means of modern transport. Our research, therefore, sought to evaluate the CleanPhone (Glissner), a UV-C light mobile phone sanitiser. Using 148 mobile phones, we concluded that the CleanPhone UV-C device is an ultra-rapid, 10 s, powerful germicidal phone sanitiser. The future of infection control is technology driven, and our public health and biosecurity authorities would benefit from such technological advances.

## Figures and Tables

**Figure 1 microorganisms-11-01876-f001:**
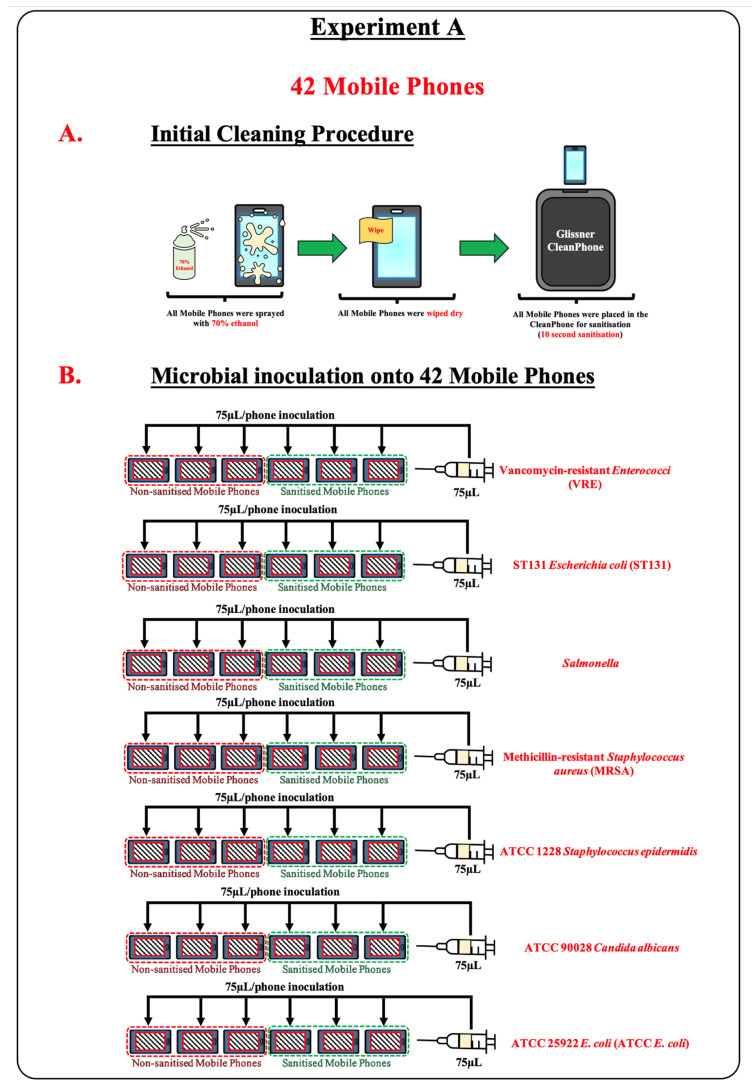
Overview of the general procedure for Experiment A.

**Figure 2 microorganisms-11-01876-f002:**
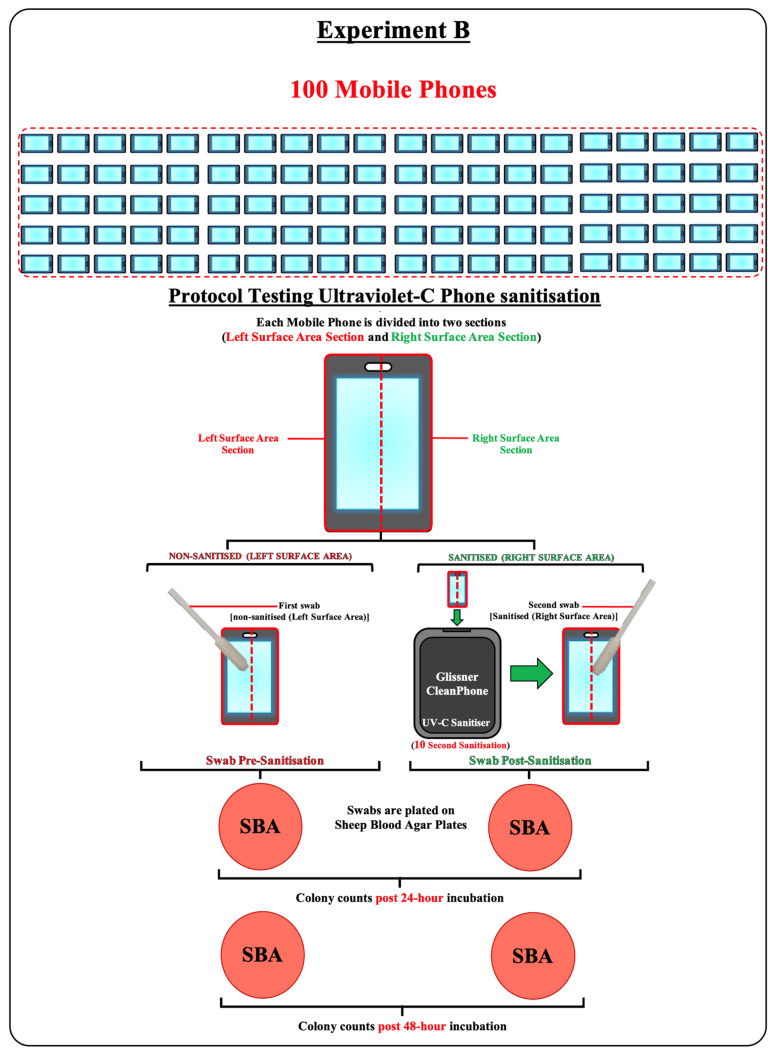
Overview of the general procedure for Experiment B.

**Figure 3 microorganisms-11-01876-f003:**
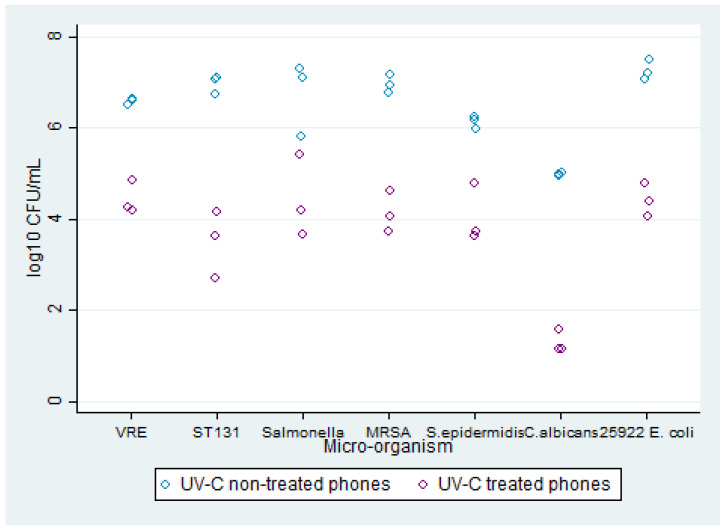
Growth concentration (log_10_ CFU/mL) of seven microorganisms on Columbia sheep blood agar from mobile phones untreated and treated with UV-C sterilisation using the Glissner CleanPhone UV-C phone sanitiser device.

**Figure 4 microorganisms-11-01876-f004:**
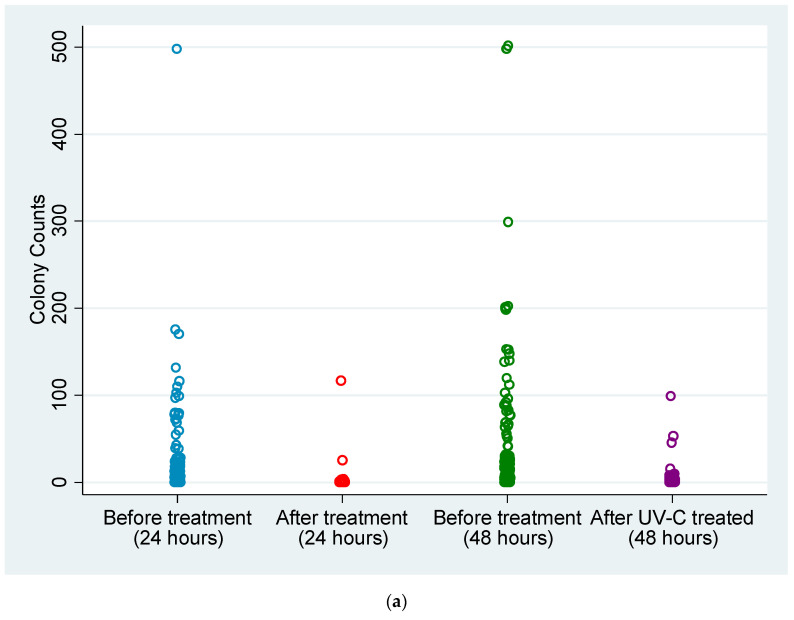
(**a**) Colony counts on Columbia sheep blood agar inoculated with swabs from 100 mobile phones before and after the sterilisation treatment using the Glissner CleanPhone UV-C phone sanitiser device. Counts was performed on all agars after 24 h and 48 h incubation. (**b**) Colony count resulting from swabs of 100 mobile phones not subject to UV-C sanitisation. (**c**) Colony count resulting from swabs of 100 mobile phones subject to UV-C sanitisation.

## Data Availability

No new data is available.

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
