# Peer review of "Ultraviolet-C-Based Mobile Phone Sanitisation for Global Public Health and Infection Control"

_microorganisms, 2023, doi:10.3390/microorganisms11081876_

Round 1
Reviewer 1 Report
The authors aimed to evaluate the germicidal capacity of the Glissner CleanPhone UV-C 86 phone sanitiser (Glissner) by applying a ten second UV-C treatment on mobile phones. The research field is interesting and the work is well written, but some improvements need to be made.
Please present the statistical analysis in more detail. Enter p-values for all results. Also insert letters to show significant differences between means obtained in the figures.
Please improve your discussion by comparing your results with other studies already carried out. Delete spaces between paragraphs in the discussion topic.
Provide a separate conclusion topic.
Review all references and adjust them according to journal guidelines.
.
Reviewer 2 Report
The manuscript entitled „ULTRAVIOLET-C BASED MOBILE PHONE SANITISATION FOR GLOBAL PUBLIC HEALTH AND INFECTION CONTROL.“ by Olsen et al. shows a way to reduce the microbial load on phone screens with a device that emits UV-C radiation. Correctly, the authors state that phones are suspected to play a role in the spreading of diseases and measures to prevent spreading events are urgently needed. However, with the submitted manuscript I have several issues that need to be addressed by major revision before the manuscript can be accepted.
Major concerns:
- Phone screens are not all the same, the technology and material might differ, as there are resistive as well as capacitive touchscreens on the market. Material properties might influence the efficacy in recovering the bacteria from the surface. Do the authors have any information on which kind of phones were used and how many of the phones used which touchscreen technology?
- The manuscript has 21 citations out of which in 7 Michael Olsen was co-author. I understand that pre-existing own work is always the foundation of further own research. However, the authors should include more references from other researchers that support the self-citations.
- In line 20 the authors use the term “Ultraviolet-C”. I would rather suggest to either write ultraviolet-c radiation or ultraviolet-c light
- In lines 40-41 the authors state “However, a large volume of scientific 40 publications…” and then there is only one citation. This is not “a large volume”. Please either use more scientific evidence or rephrase this sentence.
- In line 89 the authors state “The performance of Glissner was evaluated…”. To me as a reader it is not clear, what Glissner is. Please clarify: it it a device? The name of a company? I would suggest using the name of the device followed by the supplier. Furthermore, I would like to see the UV-C light emitting device explained in more detail. Which wavelength does the light source use and what intensity? Without this information, all the following data the authors collected are not comparable to other studies that used UV-C light.
- In lines 99-100, it is mentioned which organisms were used. However, for Enterococcus, Salmonella and S. aureus I miss the strain designations. Who was the supplier of the organisms or where were they derived from?
- Lines 104-105: Why did the authors use RPMI broth for cultivation of C. albicans? Typically, this medium is not used for cultivation of yeasts but rather for eukaryotic tissue culture
- In line 114, the authors state that they inoculated with 75µl, but give no information with what exactly was inoculated. I suspect that the authors used the bacterial culture. If so, why did the authors not adjust the optical density of the inoculum to a defined optical density?
- Did the authors also check on the CFU/ml of the inoculum? When recovering microorganism from surfaces, it is also crucial to know the amount of microorganisms that were initially applied.
- A question concerning experiment B: Was the area that was swabbed always the same?
- In lines 186 to 191, there is a decrease in the average colony count after 48h. This seems odd to me, as I guess that this decrease is due to the exclusion of all samples that showed no growth. In my opinion, all agar plates should be included in the evaluation.
- In the discussion, the values of the microbial reduction differ from the ones given in the results section. Can the authors comment on that? I think this is confusing to the reader.
- Some general remarks concerning the discussion: I would suggest putting the research more into context. How did the method perform compared to other phone disinfection strategies, for example?
- The figures are composed of a lot of – in my opinion – unnecessary information. I do not think it is important to show all agar plates or 100 phones for example. Make clear what’s the concept and do not repeat information.
Minor concerns:
- The authors use the term micro-organism. I would rather suggest writing microorganism.

Round 2
Reviewer 1 Report
The paper has improved.
.
Reviewer 2 Report
I thank the authors for addressing all comments - I think they improved the quality of the manuscript.